# An Automatic Nuclei Image Segmentation Based on Multi-Scale Split-Attention U-Net

**Qing Xu**                                                                    xq14183925@gmail.com
**Wenting Duan**                                                               wduan@lincoln.ac.uk
*School of Computer Science, University of Lincoln, Lincoln, United Kingdom*

**Editor:**

## Abstract

Nuclei segmentation is an important step in the task of medical image analysis. Nowadays, deep learning techniques based on Convolutional Neural Networks (CNNs) have become prevalent methods in nuclei segmentation. In this paper, we propose a network called Multi-scale Split-Attention U-Net (MSAU-Net) for further improving the performance of cell segmentation. MSAU-Net is based on U-Net architecture and the original blocks used to down-sampling and up-sampling paths are replaced with Multi-scale Split-Attention blocks for capturing independent semantic information of nuclei images. A public microscopy image dataset from 2018 Data Science Bowl grand challenge is selected to train and evaluate MSAU-Net. By running trained models on the test set, our model reaches average Intersection over Union (IoU) of 0.851, which is better than other prominent models, especially 4.8 percent higher than the original U-Net. For other evaluation metrics including accuracy, precision, recall and F1-score, MSAU-Net shows better performance in the most of indicators. The outstanding result reveals that our proposed model presents a promising nuclei segmentation method for the microscopy image analysis.

**Keywords:** Nuclei Segmentation, Convolutional Neural Network, Multi-scale Split Attention.

## 1. Introduction

Automatic nuclei segmentation plays an essential role in the microscopy image analysis. Traditional segmentation algorithms are based on edge detection, thresholding, morphology, pixel energy or distances between each cell, such as watershed (Meyer, 1994), Otsu (Otsu, 1979) and snake algorithm (Kass et al., 1988). Each method has different parameters which can be adjusted manually to adapt various requirements. However, achieving cell segmentation using these methods needs to combine with other image processing tools including grayscale or histogram equalization so that the final model often exists a number of parameters which are difficult to be optimized (Riccio et al., 2018). In addition, deep learning techniques based on Convolutional Neural Networks (CNNs) have led to a significant breakthrough in the image classification and semantic segmentation. U-Net, proposed by Ronneberger et al. (2015), removes the fully connected layer and adds more feature channels in the up-sampling part so that the semantic information in the network can be broadcasted to higher resolution layers. It indicates a promising performance in the cell segmentation. Zhou et al. (2018) introduced a nested U-Net (Unet++) for medical image segmentation. The model adds a series of nested and skip pathways in original U-Net, which enhances the ability of feature sharing between each sub-network. Jha et al. (2020)

constructed a DoubleU-Net architecture that adds another U-Net at the bottom of U-Net in order to obtain more semantic information from images. The model also applies Atrous Spatial Pyramid Pooling (ASPP) to acquire contextual information. By experiments on a nuclei image dataset, DoubleU-Net illustrates the better segmentation result than U-Net and nested U-Net. However, the performance of these methods is affected by image acquisition quality. For example, some of pathological images are blurred or contain noises. Other situations include uneven illumination, low image contrast between foreground and background, touching cells, large variations in nuclei sizes and shapes (Weigert et al., 2020). Therefore, a more robust approach is urgently needed.

In this paper, we propose a U-shape deep neural network called Multi-scale Split-Attention U-Net (MSAU-Net) for rapid and automatic segmentation of cells in microscopy images. The network consists of two parts: encoder module and decoder module. Both modules adopt channel-wise attention mechanism with multi-scale convolutions so that the network can obtain multi-scale features from input images and enhance its receptive field. Experimental analysis is conducted with a public dataset, which was acquired from the Data Science Bowl grand challenge in 2018. The proposed MSAU-Net model demonstrates an outstanding segmentation result in terms of the accuracy, precision, recall, F1-score and Intersection over Union, which presents a potential cell segmentation tool for medical image analysis.

## 2. Method

### 2.1. MSAU-Net Architecture

The first U-Net uses simply convolutional block as the encoder, so that it has limited ability to extract semantic features from images. In order to capture more semantic information and enhance the robustness of U-Net, we propose a novel MSAU-Net framework for nuclei segmentation. The model replaces the original convolutional layer of U-Net with our presented Multi-scale Split-Attention block, which is inspired by Split-Attention block (Zhang et al., 2020). The comparison between Split-Attention block and our proposed block is illustrated in Figure 1.

The Split Attention block involves two components: feature group and split attention operations. Although more cardinal and split groups can improve the ability of feature extraction for the network, the number of parameters of the network will also grow rapidly. To reduce the number of parameters with larger groups, our network only retains one cardinal group. A set of transformations $\{F_1, F_2, ...F_r\}$ is applied into every group. Each transformation is a $1 \times 1$ convolution followed by $k \times k$ convolution, which is defined as:

$$k = 2r + 1, \quad r \in 1, 2, ...N \tag{1}$$

These multi-scale convolutions can extract features with different scales from input images. The middle representation of each group is $E_i = F_i(X), \quad i \in \{1, 2, \ldots r\}$. In addition, the cardinal group is represented as:

$$\hat{E} = \sum_{j=1}^{r} U_j, \quad \hat{E} \in \mathbb{R}^{H \times W \times C} \tag{2}$$

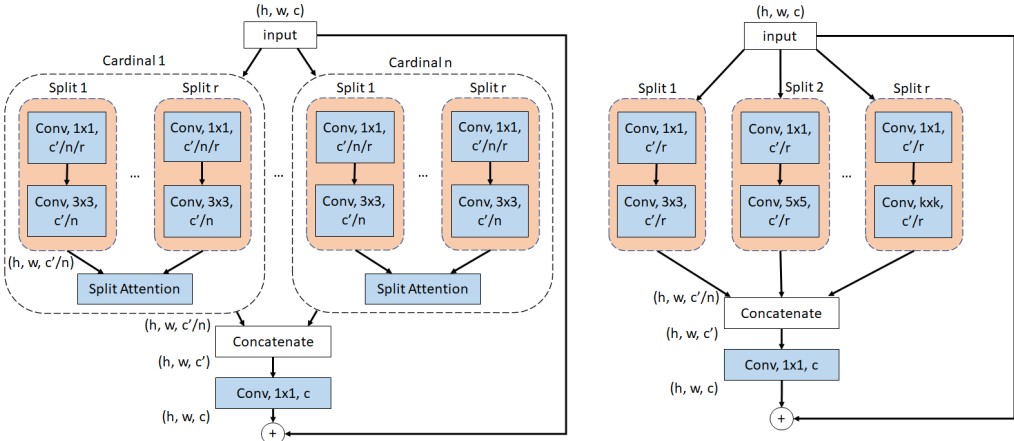

Figure 1: Comparing our block (right) with Split-Attention block (left)

Where H, W and C are the sizes of output features. The global average pooling is able to collect the global context information of embedded channel-wise statistics through spatial dimensions. The information in the c-th spatial dimension is calculated as:

$$S_c = \frac{1}{H \times W} \sum_{i=1}^{H} \sum_{j=1}^{W} \hat{E}_c(i,j), \quad S \in \mathbb{R}^C \tag{3}$$

The channel-wise soft attention is used to aggregate a weighted combination of cardinal group representation, where each channel of the feature map is generated using a weighted fusion over splits. Then the c-th channel is defined as:

$$V_i = a_i(c) E_i \tag{4}$$

Where $a_i(c)$ indicates a (soft) assignment weight provided by:

$$a_i^n(c) = \begin{cases} \frac{exp(\mathcal{G}_i^c(S))}{\sum_{j=1}^{r} exp(\mathcal{G}_i^c(S))} & r > 1 \\ \frac{1}{1+exp(\mathcal{G}_i^c(S))} & r = 1 \end{cases} \tag{5}$$

According to the globally contextual information $S$, $\mathcal{G}_i^c$ controls the weight of every split group for the k-th channel, which is calculated by two fully connected layers with ReLU activation. All of channels will be concatenated as:

$$V = Concat\{V^1, V^2, \dots V^i\} \tag{6}$$

As a result, a shortcut connection is used to produce the final output of the Multi-scale Split-Attention block: $Y = V + X$, if the input feature map shares the same shape with the output feature map. When applying a transformation $\mathcal{T}$, such as the stride convolution, on the shortcut connection, the output of the block is: $Y = V + \mathcal{T}(X)$. The full MSAU-Net model has been provided in Figure 2. The left side of MSAU-Net is referred to as

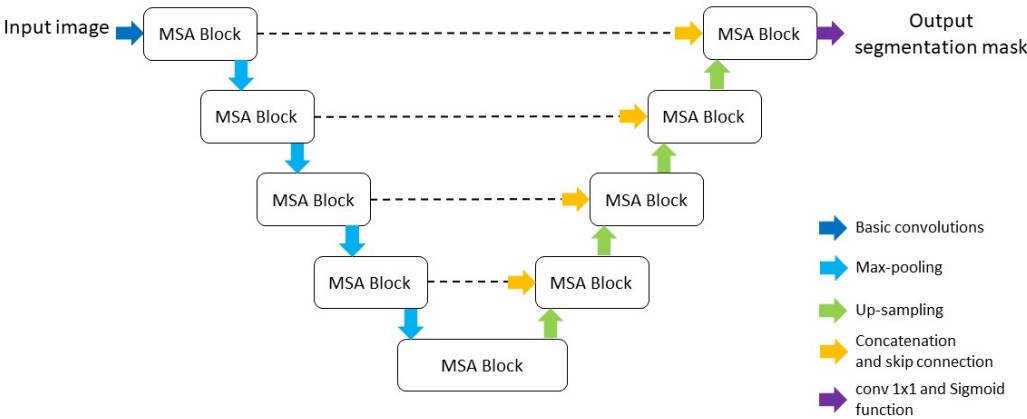

Figure 2: Presentation of our designed MSAU-Net model

encoder and the right side as decoder. The Split-Attention blocks are used to extract complex features from the input image. A $2 \times 2$ max pooling with stride 2 is executed after each block for down-sampling. After performing down-sampling 4 times, MSAU-Net will start decoding these features. Every feature representation learned from each block of the encoder is copied into the corresponding block of decoder and then concatenates them with the output of up-sampling layer. The bilinear interpolation algorithm uses all nearby pixels to calculate the value of unknown pixel and is applied on the up-sampling layer to recover the original size of image gradually. The fused feature representation after concatenation is propagated by skip connection. On the other hand, a $1 \times 1$ convolution is used to transfer every 16-channel feature map to the required number of predicted classes in the final layer.

### 2.2. Loss Function

In deep learning, loss function quantifies the difference between the result of prediction and label. Cross-Entropy (CE) loss is a common loss function used for training deep neural networks. However, it cannot perform well in the imbalanced data (Jadon, 2020). To avoid this issue, we train the model using a combined loss function using Logarithmic Dice Loss and Focal Loss (Pan et al., 2019), which is calculated as:

$$Loss\left(\hat{y}, y\right) = \alpha \times \left(-\log\left(1 - \frac{2\sum \hat{y}y + s}{\sum \hat{y} + \sum_i y + s}\right)\right) + (1 - \alpha) \times \log(\hat{y}) \times (1 - \hat{y})^{\gamma} \quad (7)$$

where $\hat{y} \equiv \{\hat{y}_i\}$, $\hat{y}_i \in [0, 1]$ is the prediction value for the i-th pixel, $y \equiv \{y_i\}$, $y_i \in 0, 1$ is the corresponding ground truth point, and s is a smoothing scalar. We set $s$ as 1 to ensure that the function is defined in edge case scenarios such as $\hat{y}_i = y_i = 0$. $\gamma$ is used to adjust the reduction rate. $\alpha$ value is optimised by F1-score. It determines the weight of Logarithmic Dice Loss and Focal Loss. The aim of Logarithmic Dice Loss can enhance the penalty for the lower prediction of the dice coefficient. It has been demonstrated to improve the performance of the model over cross entropy loss in semantic segmentation

tasks (Novikov et al., 2018). On the other hand, Focal Loss (Lin et al., 2017) improves the weight of fuzzy foreground so that some of cells in the microscopy image can be segmented easily. The combined loss function considers the benefits of both Logarithmic Dice Loss and Focal Loss functions.

## 3. Experiments

### 3.1. Dataset

Data Science Bowl, in 2018, launched a competition with a requirement to design an efficient algorithm for automatic segmentation and detection of nuclei in microscopy images. There are 670 nuclear images and corresponding pixel-level segmentation masks in the dataset of 2018 Data Science Bowl grand challenges, where 600 cell images and their segmentation masks are set as the training set and remaining 70 samples as the testing set. At the training stage, we randomly select 85% samples of the training set to train the model and the rest of 15% samples are used for validation.

### 3.2. Training Strategy

In this section, an algorithm that uses the proposed MSAU-Net architecture is designed to achieve nuclei segmentation. We set 2 split groups within the cardinal group. The numbers of kernel in the first five convolutional layers of the encoder were set to 16, 32, 64, 128 and 256, and remaining convolutional layers in the decoder were set to 128, 64, 32, 16 and 1. The size of input images is 256×256 pixels. An Adam optimization technique (Kingma and Ba, 2014) is used to minimize the loss function. The learning rate for training the model is determined by a sequential search in a series of values 0.05,0.01,0.005,0.001,0.0005,0.0001 and choosing one that has the lowest validation loss. In order to guarantee the model can be converged completely, the batch size and epochs are set to 8 and 200 in training. The IoU value is calculated during the training and validation, which is used for monitoring the performance of the model. For comparison, we also train U-Net, Unet++ and DoubleU-Net models which are prominent methods in terms of nuclei segmentation. The loss function, optimizer and training strategy of these models are also the same as MSAU-Net. In addition, all models will also be trained using cross-entropy loss function to compare with the combined loss function. To prevent the model to be overfitting, data augmentation approaches, such as horizontal flip and brightness transformations, are adopted in experiments. All built models were implemented using PyTorch 1.5 deep learning library, Python 3.7 and NumPy 1.14. Training and validating models were performed on a Nvidia RTX2060 GPU, 16GB RAM and an Intel AMD Ryzen 3500X CPU.

### 3.3. Results and Discussion

For MSAU-Net, the learning curve using CE loss is presented in Figure 3. It can be observed that our proposed model converges after approximately 60 epochs. At the end of training, the best validation loss and IoU reached 0.0623 and 0.7850, respectively. Figure 4 indicates the training result of MSAU-Net using the combined loss function. It can be demonstrated that this model converges faster than the model trained with CE loss and the final validated IoU is higher. For the comparison between the performance of MSAU-Net and other models

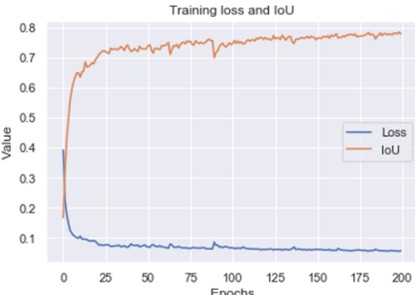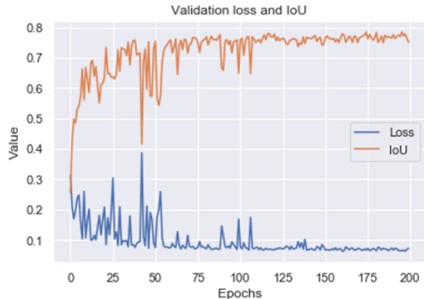

Figure 3: Results of training (left side) and validation (right side) trained on MSAU-Net with CE loss.

on the testing dataset, we calculate the average of accuracy, precision, recall, F1-score and IoU to evaluate all models.

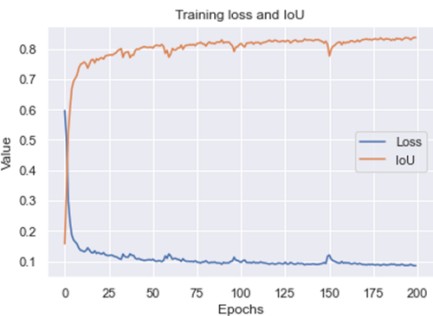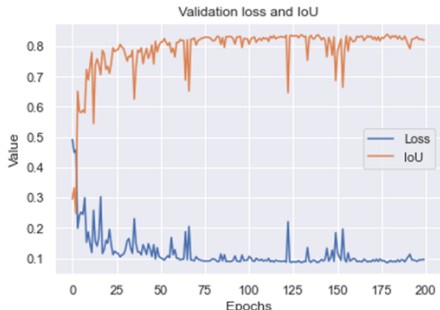

Figure 4: Results of training (left side) and validation (right side) trained on MSAU-Net with the combined loss.

Table 1: Comparison of model performance (CE) on test dataset in terms of the evaluation metrics in mean and standard deviation

| Method | Accuracy | Precision | Recall | F1-score | IoU |
|---|---|---|---|---|---|
| U-Net | 0.934±0.071 | 0.952±0.084 | 0.838±0.110 | 0.885±0.081 | 0.803±0.120 |
| Unet++ | 0.941±0.068 | 0.951±0.049 | 0.872±0.113 | 0.905±0.075 | 0.835±0.116 |
| DoubleU-Net | 0.941±0.068 | **0.957±0.039** | 0.865±0.131 | 0.903±0.089 | 0.833±0.129 |
| MSAU-Net | **0.944±0.066** | 0.938±0.069 | **0.893±0.122** | **0.907±0.104** | **0.842±0.128** |

Table 1 shows the comparison between each model trained with CE loss. In the test set, MSAU-Net achieves higher scores in the most of indicators.

Table 2: Comparison of model performance (combined loss) on test dataset in terms of the evaluation metrics in mean and standard deviation

| Method | Accuracy | Precision | Recall | F1-score | IoU |
|---|---|---|---|---|---|
| U-Net | 0.942±0.068 | 0.925±0.077 | 0.902±0.094 | 0.910±0.070 | 0.841±0.107 |
| Unet++ | 0.942±0.068 | **0.948±0.068** | 0.884±0.105 | 0.911±0.072 | 0.844±0.111 |
| DoubleU-Net | 0.943±0.066 | 0.939±0.058 | 0.893±0.094 | 0.913±0.094 | 0.846±0.102 |
| MSAU-Net | **0.944±0.066** | 0.927±0.066 | **0.912±0.098** | **0.916±0.063** | **0.851±0.098** |

In Table 2, it can be indicated that all models can get benefits from the combined loss. Where MSAU-Net achieves an average IoU of 0.851 which is better than other models, especially 1% higher than U-Net. The average precision of MSAU-Net (0.927) is lower than Unet++ (0.948) and DoubleU-Net (0.939), but MSAU-Net shows the highest average recall among all models. Also, the average accuracy and F1-score of MSAU-Net are greater than U-Net, Unet++ and DoubleU-Net. In addition, MSAU-Net spends 22s in inference, which is almost the same as other models and indicates potential feasibility for the segmentation of full-scale microscopy images. Overall, our proposed model shows better performance than these previous models in the aspect of nuclei segmentation.

At the next stage, to visualise more detailed results of the segmentation between the MSAU-Net and U-Net, we assess samples of segmentation results with different sizes of cells from the test set. The comparison of the qualitative results between ground truth images, U-Net and MSAU-Net is provided in Figure 5. From the first row of the image, it can be claimed that both models have a satisfied segmentation result if there is a few of cells in the microscopy image and these nuclei are easy to be discriminated from the background. However, for more complex images, such as the third and final rows of the image, it can be demonstrated that other models cannot completely solve nuclei images with different shapes. Especially, U-Net has a very limited ability to detect unregular cells but MSAU-Net can properly segment them from the background. As a result, the segmentation masks produced by our proposed model performs well for recognising different sizes of nuclei in the microscopy image.

MSAU-Net shows an enhanced performance in nuclei segmentation compared with the classical U-Net. However, it still can be optimized in several aspects. Firstly, both encoder and decoder of our proposed model adopts the same number of blocks as the traditional U-Net. However, there is no evidence to prove that these blocks are able to extract enough semantic information from cells. It is worth to explore MSAU-Net with deeper encoder and decoder. Moreover, because convolutions with different sizes can extract multi-scale semantic information from images, splitting more groups in the cardinal group probably can improve the performance of MSAU-Net. For this reason, further experiments will explore the impact of using different numbers of split groups in terms of the network performance. On the other hand, in the decoder of MSAU-Net, each feature map after down-sampling

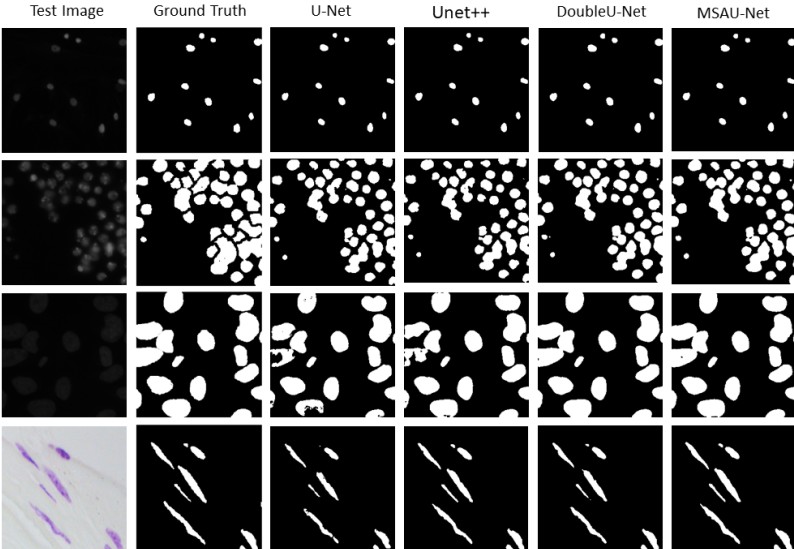

Figure 5: Segmentation comparisons between MSAU-Net and other models.

feed into one layer of the decoder. However, it is hard to confirm if the defined feature can certainly help decoder recover images. In order to improve the efficiency of the decoder, further research can try to expand the receptive field of the decoder, which is able to receive both low-level and high-level semantic information of the input image.

## 4. Conclusions

In this paper, we propose an enhanced version of U-Net called MSAU-Net for implementing the microscopy image segmentation task. The MSAU-Net replaces the original convolutional layers of U-Net with Multi-scale Split-Attention blocks that apply channel-wise attention mechanism with multi-scale convolutions. The evaluation result shows that our presented model demonstrates a higher segmentation performance than U-Net and other popular models in terms of the accuracy, recall, F1-score and Intersection over Union. In the further work, we will continue to optimize the number of parameters in the MSAU-Net architecture in order to simplify the framework and reduce the time of training and evaluation. To improve the performance of nuclei segmentation, more image data with various shapes need to be collected in the future for capturing different characters of cells. At the same time, we will continue the explore the usability of the MSAU-Net in different medical image segmentation tasks, such as brain or lung segmentation from different modalities (MRI or CT).

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
