# OpenReview forum: "An Automatic Nuclei Image Segmentation Based on Multi-Scale Split-Attention U-Net"
_MICCAI.org/2021/Workshop/COMPAY — COMPAY 2021_

### Official Review · Reviewer_ZxSX · 2021-08-16
**Marginal improvement on a decade-old task**

**Rating:** 6
**Confidence:** 4

**Review:**

In this study, the authors propose a novel variant of a U-Net for nucleus segmentation in pathology images. The technical side of the paper is interesting and has some innovative aspects. However, the application and validation of the method is rather limited. The authors use only a single public dataset to benchmark their method. Considering the fact that nucleus segmentation has been studied for decades, that thousands of methods are available and that a wide range of image data are publicly available, I conclude that a more extensive validation is needed to really convince users to use this new method.

---

### Official Review · Reviewer_Gbei · 2021-08-17
**Good idea, but marginal progress**

**Rating:** 6
**Confidence:** 4

**Review:**

The authors present a novel nucleus segmentation algorithm. They describe their method more or less clearly and show quantitative results based on a Kaggle Challenge data set.

The authors seem to follow all standard procedures for a DL paper (problem, dataset, network, experiment, result, some images). However, it is unclear to me what the contribution is: On the one hand, their new network block seems to be just a simplified version of an already existing architecture (Figure 1). On the other hand, their results do not seem to be significant at all. These are pixel-wise segmentation metrics, and an increased F1 score in the third digit after the comma does not seem to be significant.

Nucleus segmentation is still a challenging problem, especially for touching cells (to separate them) or complex shapes. Here, the authors should include a measure if their "improved" segmentation accuracy on pixel basis helps for these challenging problems.

Further, access to the source code (after publication) would be good to enable others to compare to their method, or use MSAU-Nets

Minor comments:
- Page 2, explain the acronym MSAU-Net at first usage
- make all "et al." italic
- Page 2 "where r stands for the number of split groups" seems to be out of context here?
- Page 2: A set of transformations... IS applied (not are)
- Page 3: In equationS (2)
- Page 4: loss function quantises -> quantifies
- Page 4: "alpha value is optimised by F1-score" What does that mean?
- Section 3.1: Please cite the dataset
- Page 6: "percision"
- Page7: "closed"

In general, the authors should review the whole manuscript for typos and formatting issues (e.g. formulas in text) to be consistent throught the document.

---

### Decision · Program_Chairs · 2021-08-25

Accept